# The Roles of microRNAs in the Cardiovascular System

**DOI:** 10.3390/ijms241814277

**Published:** 2023-09-19

**Authors:** Francesco Nappi, Sanjeet Singh Avtaar Singh, Vikram Jitendra, Almothana Alzamil, Thibaut Schoell

**Affiliations:** 1Department of Cardiac Surgery, Centre Cardiologique du Nord, 93200 Saint-Denis, France; almothana.md@gmail.com (A.A.); tiboschoell@hotmail.com (T.S.); 2Department of Cardiothoracic Surgery, Royal Infirmary of Edinburgh, Edinburgh EH16 4SA, UK; sanjeet.singh@glasgow.ac.uk; 3Department of Cardiothoracic Surgery, Aberdeen Royal Infirmary, Aberdeen AB25 2ZN, UK; vikram.jitendra@nhs.scot

**Keywords:** microRNA, miRNA biomarker, microRNA functions, acute coronary syndrome, coronary artery disease

## Abstract

The discovery of miRNAs and their role in disease represent a significant breakthrough that has stimulated and propelled research on miRNAs as targets for diagnosis and therapy. Cardiovascular disease is an area where the restrictions of early diagnosis and conventional pharmacotherapy are evident and deserve attention. Therefore, miRNA-based drugs have significant potential for development. Research and its application can make considerable progress, as seen in preclinical and clinical trials. The use of miRNAs is still experimental but has a promising role in diagnosing and predicting a variety of acute coronary syndrome presentations. Its use, either alone or in combination with currently available biomarkers, might be adopted soon, particularly if there is diagnostic ambiguity. In this review, we examine the current understanding of miRNAs as possible targets for diagnosis and treatment in the cardiovascular system. We report on recent advances in recognising and characterising miRNAs with a focus on clinical translation. The latest challenges and perspectives towards clinical application are discussed.

## 1. Introduction

The identification of a small noncoding RNA with a regulatory role in gene expression at the post-transcriptional level dates back thirty years in Caenorhabditis elegans [1,2]. In higher eukaryotes, continued progress in molecular biology led to the identification of multiple microRNAs or miRNAs. Discovering their widespread regulation of mammalian mRNAs was a successful gamble [3]. However, the issue of how many miRNAs are expressed in humans remains unresolved to this day. Since 1973, various annotations regarding human miRNAs have been reported in mirBase 22.1 [4]. It has been observed that many miRNAs fail to meet strict criteria such as expression, sequence constraints, or evidence of productive processing of the precursors. This issue has led to a notable fluctuation in the number of effective miRNAs in humans, ranging from 556 (mir-GeneDB 2.0) [5] to 758 [6]. Another significant factor is the proportion of miRNAs that are functionally relevant and require adequate expression levels in the tissue. Consequently, the proportion of functional miRNAs may decrease, impacting their efficacy. It is possible to hypothesise optimistically, albeit based on provisional certainties, that up to 150 miRNAs play a critical role in the cardiovascular system. Validation has been conducted on the analysis of 30–35 miRNAs. Comprehensive in vivo experimental models have also highlighted the clinical significance of many of these candidates and their potential for diagnosis and treatment (Figure 1). Furthermore, equally promising predictions concern many other candidate miRNAs, implicated in the diagnosis and therapy of cardiovascular diseases.

Although polymorphisms in miRNA biogenesis factors, miRNA genes, or miRNA response elements (MREs) have been identified in a genome-wide association study (GWAS) [49], their pathophysiological consequences have only been resolved in a select few cases. Conversely, numerous miRNAs that are deregulated or modified in disease can be highlighted. The fundamental query is whether the deregulation of a microRNA causes a disease or simply indicates it. miRNAs that play a pivotal role in disease pathophysiological processes are frequently linked with increased baseline steady-state expression, marked dysregulation during disease, and a proclivity to be present together in cells/tissues, as demonstrated in Figure 1. For instance, individuals with a transverse aortic constriction (TAC) model of ventricular pressure overload possess miR-21-5p as the prevalent miRNA in cardiac macrophages. This miRNA is also upregulated seven-fold in the myocardium [50]. Another example involves miR-29b-3p, which is highly expressed in cardiac myocytes and upregulated, by approximately three-fold in the presence of TAC [11]. Several of the 30–35 miRNAs that have substantially demonstrated critical cardiovascular roles induce distinct pathophysiological effects in the myocardium or vasculature following manipulation (Figure 2A). Some of these have a pathophysiological action dependent on the involvement of signalling pathways, leading to the secretion of protein factors (Figure 2B), while for others, the action is specifically exosome-mediated as they are themselves secreted in extracellular vesicles (Figure 3C). The expansion of knowledge on miRNAs has led to a rise in therapeutic research on these molecules in the myocardium and vascular system [51]. This new direction in cardiovascular research is depicted in Figure 1 and in Figure 2 [3,7,8,9,10,11,12,13,14,15,16,17,18,19,20,21,22,23,24,25,26,27,28,29,30,31,32,33,34,35,36,37,38,39,40,41,42,43,44,45,46,47,48,52,53,54,55,56,57,58].

The purpose of this review is to emphasise the cardiovascular function of some of these potential miRNA candidates and their possible clinical usage in the diagnosis and treatment of cardiovascular ailments. Although there are limitations to our word count, we can still provide an overview of miRNAs in the cardiovascular system. Specifically, we will focus on their role as circulating biomarkers in acute coronary syndromes. We will not be able to discuss all miRNAs shown in Figure 1 in detail. We believe that these data could become a basis for a thorough evaluation of the role of miRNAs in the cardiovascular system. It could also help physicians discuss potential new therapeutic and diagnostic approaches with their patients, including the benefits and expectations.

## 2. microRNAs’ Pathophysiologic Surroundings: Insight on Biogenesis, Stability, and Strand Bias of microRNAs

miRNAs consist of noncoding RNA sequences that come together to form ~22–24 nucleotides. Their primary functions are to fine-tune gene expression, control tissue growth, and regulate homeostasis [59,60]. In terms of regulating gene expression, miRNAs not only affect the cell of origin but also play a vital role in governing intercellular communication. During the initial stage, miRNAs are produced from pre-miRNA, which is synthesised by the RNA polymerase II enzyme on a DNA substrate. Subsequently, a range of enzymes are involved in modifying the original structure of the miRNA and transporting it to the cytoplasm. The mature miRNA then forms the RNA-induced silencing complex (RISC) with other proteins, including Dicer, Argonaute 2, and the transactivation response RNA-binding protein [61,62]. Finally, miRNAs bind to the 3′ untranslated regions of target messenger RNAs via their eight-nucleotide seed sequence. Semi-complementarity between mature miRNA and target mRNA facilitates translational repression, whereas complete complementarity between them supports mRNA degradation [63,64,65].

Although accessory pairing beyond the seed sequence may enhance target recognition, it seems that only a few microRNAs rely on such interactions [66,67,68]. miRNA response elements, which serve as miRNA target sites in mRNA, are mainly situated within the 3′-UTR and, to a lesser degree, in the 5′-UTR or coding regions [66,69]. Unlike the function of lncRNAs or circRNAs, for which various mechanisms of action have been identified, miRNAs have two distinct activities. The primary function is to promote the degradation, while the secondary function is to induce the translational silencing of target miRNAs [66].

The miRNA folder is complemented with nongenetic variations, known as isomiRs, which arise from diverse miRNA processing, nucleotide addition, or editing [70,71]. Many cardiovascular isomiRs have been identified [72,73], with varying levels observed in disease [72]. Distinct variations and targetomes were noted for R-isomers of miR-487b-3p and miR-411-5p [74,75]. The generation of a microRNA relates to its enzymatic degradation at the end of its life cycle. Most miRNAs have been demonstrated to possess a significantly longer half-life than mRNAs, with significant variation depending on the strand and sequence of the miRNA, the type of cell, and transactional factors [76,77] (see Figure 1A). Furthermore, targets of miRNAs have been identified. Although the mechanistic features of target-directed miRNA degradation (TDMD) have been established [63,78,79], and the in vivo significance of TDMD has been well demonstrated [80], it remains challenging to identify mRNAs that are involved in TDMD.

miRNAs are vital contributors to embryonic growth, the regulation of physiology, and the functioning of the cardiovascular system [81,82,83]. Changes in the cellular expression patterns of miRNAs are observed in different cardiovascular disorders like acute myocardial infarction (AMI), heart failure, and cardiac hypertrophy. Moreover, hypertension, obesity, hyperlipidaemia, and diabetes promote elevated levels of miRNAs [83,84,85,86]. Various experimental models of atherosclerosis were evaluated with a focus on miRNAs’ functionality. The analysis of atherogenesis in vitro and in vivo has indicated a significant role for various miRNAs. Specifically, phenomena such as inflammation, the activation of endothelial cells, angiogenesis, the proliferation and migration of vascular smooth muscle cells, and the formation of neointima have been associated with increased miRNA expression [86,87].

The process of actively and selectively secreting miRNAs in living cells has been extensively explained, ascertaining its important role in cell–tissue communication [88,89,90]. The transportation of secreted miRNAs is facilitated by extracellular vectors, dependent on specific molecular components, including Argonaute 2, nucleophosmin, and lipoproteins. Again, miRNAs can also be upregulated into extracellular macrovesicles and subsequently transferred into recipient cells where they alter gene expression [88,91,92]. These observations suggest that, unlike mRNA turnover, miRNAs persist in a stable state for the duration of their circulation, with optimal protection against endogenous ribonuclease activity [88,89,92]. These findings reinforced the idea that circulating miRNAs could function as potential biomarkers, capable of detecting the various forms of acute coronary syndrome (ACS). Many reports have evaluated the diagnostic potential of miRNAs in these clinical contexts [3,52,70,78,79,93,94,95,96] (Figure 3).

## 3. Searching Roles of microRNAs in the Cardiovascular System

Several studies have shown a significant increase in miR-21-5p expression in the compromised human cardiac muscle. This association of miR-21-5p with fibrosis has indicated its increased expression in renal and pulmonary diseases where fibrotic tissue degradation is a common symptom. In animal models, miR-21 inhibitors have exhibited efficacy in preventing cardiac fibrosis or neointimal formation [12]. Although a global absence of miR-21-5p may not be widely recognised, the recurring effects of its inhibitors are demonstrated through a genetic knockout of miR-21 in nonmyocyte cells. This highlights their pivotal function in those cells. Cardiac macrophages and fibroblasts exhibit significantly elevated levels of miR-21-5p. miR-21-5p deficiency in macrophages of mice resulted in resistance to transverse aortic constriction induced by structural and functional phenotypes, linked to reduced inflammatory phenomena. Furthermore, administering anti-miR-21 to pigs post-ischaemia–reperfusion showed enhanced cardiac function and reduced inflammation. The results suggest that miR-21-5p plays a significant profibrotic and pro-inflammatory role in the myocardium. As a result, a phase II study is currently assessing the effectiveness of locked nucleic acid (LNA)–anti-miR-21 for treating fibrotic renal disease.

The miR-29 family comprises four almost identical variants, which are believed to regulate collagen and other matrix proteins. Therefore, it has been identified as a potential target for antifibrotic therapies. A notable study carried out by von Roji et al. [15] demonstrated that miR-29 mimics had a repressive effect on collagen, resulting in improved cardiac function. Moreover, the study identified the potential of miR-29 mimics in enhancing cardiovascular health.

This concept has been restated in other publications, culminating in the development of a final miR-29 mimic (MRG-201), which proves effective in treating idiopathic pulmonary fibrosis. The principle focuses on reducing collagen expression; yet, anti-miR-29b promotes the stabilisation of the vascular wall, which undergoes structural changes, leading to abdominal aortic aneurysms in mouse models [16,17]. In contrast to these findings, Sassi and colleagues observed that the inhibition, rather than elevation, of miR-29 prevents cardiac fibrosis [11]. This unexpected and surprising result surprised experts in the field, but the authors elucidated its underlying mechanism. Each of the miR-29 variants is predominantly expressed in cardiac myocytes with high levels in cardiac fibroblasts that are only recorded with prolonged cultivation. The primary pathophysiological mechanism for inducing fibrosis in fibroblasts through miR-29 is via the activation of the Wnt pathway, leading to cell hypertrophy and paracrine signalling. Therefore, inhibiting miR-29 would be an appropriate intervention in the myocardium, while elevating it could suppress fibrotic pathways in fibroblasts for patients with skin diseases.

Two separate studies [21,24] examined the increased expression of miR-92a-3p in endothelial cells and its disruption in mouse models with myocardial and vascular tissue lesions. A practical LNA anti-miR against miR-92a has been demonstrated to promote angiogenesis and tissue restoration in these models [24], which has since been validated in a study based on a porcine ischaemia–reperfusion model [21]. These data have been translated with a view to clinical application, based on the findings of a pharmacological study on anti-miR-92a (MRG-110). The cohort enrolled healthy individuals who were administered a single drug via intravenous injection. The enrolled cohort consisted of healthy individuals who received a single drug by intravenous injection [97]. Importantly, given the efficacy of anti-miR-92a after intradermal inoculation, which was also found in animal models of skin injury, a second phase I clinical study using this route of administration was designed (ClinicalTrials.gov NCT03603431; Accessed on 27 July 2018).

The upregulation of miR-155-5p has been reported in patients with cardiac inflammation or in corresponding animal models [23,26]. Bone marrow transplantation experiments in mice have suggested that the proinflammatory activity of miR-155 is linked to macrophages [91], in which NF-κB expression is increased. In contrast, an increase in miR-146a-3p expression opposes this effect [92]. Although inhibiting miR-155 improves cardiac inflammation in mice [23,26], further analysis is necessary to support initial findings that macrophage-specific miR-155 deficiency prevents arteriogenesis after vascular injury [28]. Cobomarsen, an anti-miR against miR-155, progressed to a phase I study in cutaneous T-cell lymphoma (ClinicalTrials.gov NCT02580552; Accessed on 20 October 2015) [98], but a phase II study was halted due to speculative interest in cardiovascular research. Despite several miRNA-targeting therapeutic developments being abandoned in other indications (e.g., miravirsen, RG-101, cobomarsen, and AZD4076), their impact on the cardiovascular field appears to be smaller than previously assumed. The preclinical and clinical data obtained from inhibitory oligonucleotides, even those that have been discontinued, provide valuable information for designing and performing miRNA-targeting cardiovascular therapies. This pertains to miR-17-5p, miR-21-5p, miR-29b-3p, and miR-92a-3p, which have been thoroughly examined in both the laboratory and clinic. However, it especially pertains to the previously mentioned miR-132-3p inhibitor (CDR132L), developed for the treatment of heart failure. CDR132L, presently slated for phase II trials, may become the premier microRNA-targeting medication in cardiovascular therapy. [99,100,101,102,103,104,105,106,107,108,109,110,111,112,113,114,115,116,117,118,119,120,121,122,123]

MiR-132-3p has successfully completed its preclinical phase and advanced to clinical trials, showing rapid progress in terms of efficacy, safety, and benefits. Notably, research has revealed that miR-132/-212 cluster genetic deficiency or the use of an antagomir against miR-132-3p can prevent TAC-induced pathological cardiac remodelling [25]. These findings prompted an assessment of miR-132-3p inhibition in mouse models of heart failure, wherein blood pressure overload was induced [22,25]. In another study employing a porcine heart failure model, the authors demonstrated that miR-132 persisted for a long time within cardiac tissue (with a t1/2 of 3 weeks), with an advantageous safety profile. The authors also confirmed the derepression of miR-132 targets. In pig models where myocardial infarction was induced, the use of anti-miR-132 improved cardiac function after myocardial injury. Similarly, in another pig model, the same effects were observed for preventing chronic pressure overload. The initial human study on dose escalation (phase Ib) for patients with heart failure highlighted good tolerability and the first indications of therapeutic gain [101].

## 4. microRNAs as Circulating Biomarkers in Acute Coronary Syndromes

A report from the World Health Organization, validated by the Center for Disease Control and Prevention (CDC), revealed that coronary artery disease (CAD) and acute coronary syndrome (ACS) are the primary causes of morbidity and mortality across the globe. These illnesses frequently result in complications. The classification of ACS is more detailed than the generic term, as acute myocardial infarction (AMI) presents itself through either electrocardiographic signs of ST-segment elevation or depression (STEMI/NSTEMI) or as unstable angina pectoris (UA). ACS encompasses a varied patient population, with only 50% of those admitted to the hospital due to chest pain of cardiac origin.

This interval of 1–2 h, which allows for the elimination of myocardial infarction by means of early diagnosis, is currently the most significant clinical hurdle to overcome. Thus, it is crucial to conduct successive assessments during the downtime for AMI diagnosis, during the extended monitoring of patients in emergency medical and/or cardiology centres, to drive the development of fresh rule-in and rule-out tactics for the timely identification of AMI. The advantages of utilising miRNAs as biomarkers were established in a groundbreaking study by Mitchell and their team almost 14 years ago.

The enhanced diagnostic capability of miRNAs can be attributed to their short, noncoding nature that regulates gene expression post-transcriptionally, and their remarkable stability in circulation. Extensive research has been conducted on the diagnostic efficiency of these molecules, and they have been suggested as biomarkers for diagnosing various diseases such as aortic dissection, where the plasma of patients exhibited human cytomegalovirus-encoded miRNA expression profile (Figure 4).

## 5. Insight into the Diagnostic and Prognostic Value of Circulating miRNAs in Coronary Artery Disease: Evaluation of the New Disease Diagnosis Guide for Stable Coronary Artery Disease and Acute Coronary Syndrome

Coronary artery disease is the prevailing expression of cardiovascular disease, and acute coronary syndrome is significantly associated with morbidity and mortality across a wider segment of the global population. There are various biomarkers available for the diagnosis of acute myocardial infarction and subsequently used. Among these, cardiac troponins are regularly used as biomarkers in individuals who suffer from chest pain; however, their sensitivity is lacking in the first few hours after the onset of symptoms, thus obscuring their complete diagnostic validity. For this reason, finding new biomarkers for diagnosing coronary events remains a priority in research aimed at clinical application. Current studies have indicated a significant involvement of miRNAs in promoting the generation of atherosclerotic plaque, and their expression is altered during the development of CAD. While multiple studies have indicated that several miRNAs may not be superior to troponin in diagnosing AMI, their use may still be directed towards the early diagnosis and prognosis of CAD patients. Further studies are required in this scenario. We summarised recent research investigating circulating miRNAs as new biomarkers in CAD, diagnosing AMIs, and in-stent restenosis. We also studied the involvement of miRNAs in CAD risk assessment, alongside assessing their impact on the prognosis of patients experiencing acute coronary syndrome.

### 5.1. miRNAs and Coronary Artery Disease

Disparate and somewhat incongruous findings have been documented in various reports concerning groups of patients experiencing acute coronary syndrome in the two clinical scenarios of stable angina (SA) and unstable angina. The data that have arisen indicate a somewhat nuanced significance and an absence of a substantial diagnostic value that can explain the actual validity and effectiveness of miRNAs in diagnosing the different types of ACS. The diagnostic advantages linked with analysing miRNAs as biomarkers in the prompt recognition of UA were confirmed in a seminal document from the German Centre for Cardiovascular Research a decade ago [124]. The significant, albeit inconclusive, findings derived from this study are attributed to the enrolment of a large patient cohort over three phases. Specifically, 667 miRNAs were identified during the initial screening phase, in which three patient cohorts were examined. Among them, 10 individuals presented with acute coronary syndrome in the form of unstable angina, while a further 10 presented with noncardiac chest pain (NCCP). The third group of participants included 20 healthy individuals who were matched for sex and gender with those patients who showed UA.

From the original pool of references, 25 miRNAs were identified as being associated with UA (*p* < 0.05). In addition, miR-1, miR-208a, and miR-208b were included in the second phase of replication for final identification. These elements underwent evaluation in an independent patient cohort comprising 49 UA and 48 NCCP patients. Eight miRNAs, namely miRNA-19a, miRNA-19b, miRNA-132, miRNA-140-3p, miRNA-142-5p, miRNA-150, miRNA-186, and miRNA-210, were deemed crucial for progression to the final stage because of their significant overlapping presence in patients diagnosed with UA during the initial two stages. The third phase entailed the validation and determination of the diagnostic capability of miRNAs in UA subjects. To confirm the findings, 46 UA patients and 63 NCCP patients from the first cohort were examined. The respective areas under the curve (AUC) were calculated at baseline, comparing UA patients to NCCP patients. The most significant results were achieved by MiR-186 (AUC = 0.78). The second and third most well-expressed miRNAs were miR-132 and miR-150, respectively, achieving an AUC of 0.91.

More research was conducted during the same period to evaluate circulating miRNA profiles in CAD patients compared with healthy controls. The goal was to identify useful miRNA expressions that could accurately distinguish between SA and UA pectoris. The authors screened 367 miRNAs and recorded key data for three miRNAs that exhibited higher expression in CAD patients than in those without the disease. An increase in miR-NA-337-5p was observed exclusively in SA but not in UA. However, miRNA-433 and miR-NA-485-3p were each increased in both SA and UA compared with healthy controls. However, none of the three miRNAs were found to significantly differ in relation to the clinical presentation of ACS, thus indicating their lack of specificity for SA or UA. The investigation also included 14 miRNAs previously studied, which had shown promising results. Among patients with CAD and healthy controls, seven of these miRNAs were found to be dysregulated.

The data revealed that levels of miRNA-1, miRNA-122, miRNA-126, miRNA-133a, miRNA-133b, and miRNA-199a were substantially increased in the presence of SA or UA, individually. miRNA-145 was significantly increased only in patients with UA. The statistical analysis confirmed predictive outcomes for AUC in both of the groups studied. In the SA patient cohort, three miRNAs exhibited a predictive value exceeding 0.85 in comparison to controls, including miRNA-1 with a value of 0.918, miRNA-126 with a value of 0.929, and miRNA-485-3p with a value of 0.851. In the UA cohort, the corresponding AUC values for miRNA-1, miRNA-126, and miRNA-133a were 0.92, 0.867, and 0.906, respectively, which were deemed satisfactory (>0.85). However, when the two SA and UA arms were compared, the identifiable statistical advantage in the AUC area was found to be ineffective, with the results falling short of diagnostic power.

Although the combination of the three most effective miRNAs could differentiate between patients with SA and UA and nonpathological controls with an efficiency of 90.2% and 87.2%, respectively, the discriminatory ability between the two groups with CAD was limited to a maximum of 66% [125]. The difference in findings becomes evident when comparing the study with Fichtlscherer et al.’s investigation, where the AUC area was assessed through the correlation of various clinical CAD conditions. In that study, the authors observed that miR-17, miR-92a, miR-126, miR-145, and miR-155 levels were lower in the CAD-SA group than in the healthy controls. Conversely, levels of miR-133a and miR-208a were elevated, but the reported values did not demonstrate statistical significance.

In patients with coronary artery disease, an assessment was carried out of the diagnostic potential of a cluster of miRNAs with altered expression. To begin with, miRNA profiles were used to screen 13 patients with unstable angina and 13 control group patients with chest pain who did not show atherosclerosis during percutaneous coronary intervention (PCI). The UA group was found to have 34 miRNAs that were significantly deregulated. The novel data in this study pertain to the categorisation of miRNAs either into clusters or as members of a specific family. When some of these miRNAs originated from nearby gene loci, they were considered clusters, e.g., the miR-106b/25/93 cluster and the miR-17/19b/20a/92a cluster. Instead, miR-21/family 590-5p has been categorised in the same family with other miRNAs due to the sharing of 5′ seed sequences. Seven miRNAs (miRNA-106b, miRNA-25, miRNA-92, miRNA-21, miRNA-590-5p, miRNA-126*, and miRNA-451) were chosen out of the 34 identified miRNAs based on their high levels of circulating abundance, previous findings, cluster definition, or membership in the same family. From the three distinct cohorts, 45 patients with UA and 31 patients with SA, as well as 37 controls, were identified (total *n* = 125).

Fichtlscherer et al. [126] and Ren et al. [127] initially proposed the methodical assessment of miRNA dosage separately, despite advocating considerable differences in acute coronary syndrome clinical scenarios. The best way to showcase the growing importance of miRNAs is to determine whether they were up- or downregulated. In patients with CAD caused by stable angina [128], miRNA-17, miRNA-20a, miRNA-21, and miRNA-92a were discovered to have lower expression levels, while in patients with CAD caused by unstable angina [127], increased levels were observed. As a result, all seven miRNAs demonstrated an upsurge in patients with unstable angina, compared with both stable angina and healthy controls, even after adjusting for risk factors. In this clinical scenario, it may be proposed that the miRNA-106b/25 cluster, miRNA-17/92a cluster, miRNA-21/590-5p family, miRNA-126*, and miRNA-451 have the potential to act as a biomarker for unstable angina.

Recently, Su et al. [129] report evaluated circulating miRNAs as noninvasive, accurate, and sensitive biomarkers for diagnosing coronary artery disease. They used the direct S-Poly(T)Plus method to enrol 203 patients who had CAD and 144 age-matched controls, which comprised 126 high-risk controls and 18 healthy volunteers. Ultimately, six miRNAs, namely miR-15b-5p, miR-29c-3p, miR-199a-3p, miR-320e, miR-361-5p, and miR-378b, were verified in validation set-2. This achieved a sensitivity of 92.8% and a specificity of 89.5% with an AUC of 0.971 (95% confidence interval, 0.948–0.993, *p* < 0.001) in a significant cohort for CAD diagnosis. These outcomes strengthen the efficacy of the direct S-Poly(T) Plus methodology for diagnosing CAD by targeting 12 unique miRNAs. Moreover, the process of plasma fractionation indicated that only minute quantities of miRNAs were accumulated within extracellular vesicles [129].

### 5.2. microRNAs and Acute Coronary Syndrome

Proponents of a substantial role of miRNA 208b, miRNA-499, miRNA-133a, miRNA-21, and miR-146a in the diagnosis of acute coronary syndrome have achieved robust evidence through sophisticated statistical investigations [125,130,131,132,133,134,135,136,137,138,139]. This role was formally recognised in two seminal papers in which assorted separate cardiac- and muscle-enriched miRNAs were evaluated and proposed as potential biomarkers of acute myocardial infarction [125,130]. Subsequently following the emerging evidence, in a larger study dictated to establishing the role of miRNAs in the diagnosis of MI, the levels of miRNA-208b and miRNA-499, as well as high sensitivity troponin (hs-Tn), were measured in 510 AMI patients and 87 healthy controls [131]. Davaux et al. [131] proposed that both miRNAs considerably increased in AMI patients compared with controls. miRNA-499 exhibited a statistical asset with an AUC area of 0.97 when contrasting healthy subjects with those exhibiting AMI. However, the emerging data did not contribute significantly to diagnostic accuracy compared with the determination of hs-Tn alone. If a diagnosis of myocardial infarction was mistakenly made in patients, and retrospective addition of miRNA-499 was later performed to rectify the assessment, the diagnosis did not significantly improve [131].

The diagnostic potential of miRNAs is weakened due to their relatively low sensitivity and specificity compared to cardiac troponin (cTn) measurement. A study analysed the expression of miRNA-1, miRNA-133a, miRNA-208b, miRNA-499, and cTnT in 67 patients with acute MI and 32 healthy volunteers as the control group. It was noted that all four miRNAs analysed were significantly increased within 12 h of the onset of chest pain in patients with AMI. Controls carried out on day 14 post-acute myocardial infarction revealed comparable levels of miRNA among the groups. The findings suggest that these miRNAs may serve as effective biomarkers for AMI, despite lower sensitivity and specificity compared with the measurement of cTnT [132].

A meta-analysis combined data from two previously published studies [132,133] to compare the ability of miRNAs to act as biomarkers in individuals who have experienced AMI versus those who are healthy. In total, 192 full-text articles were reviewed, with 19 articles selected for the final analysis. In 15 of these, the four miRNAs most frequently mentioned in the articles were examined in detail. miRNA-499 was analysed in eight studies, miRNA-1 in seven studies, miRNA-208b in six studies, and miRNA-133a in four studies, respectively, yielding compelling statistical evidence. The reported values for the AUC indicated that all miRNAs were established and trustworthy biomarkers for AMI. Again, miRNA-133a and miRNA-499 may remain important biomarkers for AMI, but the role of miRNA-208b needs further study [133].

For a considerable time, researchers have studied the role of circulating miRNAs within the cellular components of whole blood, with no research conducted on subcomponents. Initially, Ward et al. [140] utilised a quantitative real-time polymerase chain reaction system to explore the miRNA profiles of subcomponents of whole blood in ACS patients, with a particular focus on plasma, leukocytes, and platelets. Patient cohorts included 13 AMI patients admitted to the emergency department or undergoing PCI with electrocardiographic signs of STEMI or NSTEMI. Arterial blood samples were collected during PCI to obtain whole blood.

The protocol mandates cell-specific profiling of miRNA, including the quantification of 343 miRNAs from whole blood, plasma, peripheral blood mononuclear cells, and platelets. The discussion is substantiated by three distinct conditions. Patients with STEMI exhibited higher levels of miRNAs and were found to have increased expression of miR-25-3p, miR-221-3p, and miR-374b-5p, compared with the NSTEMI cohort. The miRNA 30d-5p noncoding RNA sequences were associated with plasma, platelets, and leukocytes in both STEMI and NSTEMI cohorts. By contrast, miRNAs 221-3p and 483-5p were solely associated with plasma and platelets in NSTEMI patients. The authors initially proposed that cell-specific miRNAs varied widely between the cohorts of patients with STEMI and NSTEMI. Similarly, this study showed that the distribution of miRNA was distinct among plasma, platelets, and leukocytes in patients with ischaemic heart disease or ACS. Furthermore, based on the reported findings, it was suggested that patients with myocardial ischaemia could be identified by their unique miRNA profiles in their circulating subcomponents [140].

Potential bias for validating the diagnostic power of miRNAs in various clinical settings of ACS, including SA, UA, and AMI, in patients admitted to the emergency department (ED) or undergoing PCI, arises from the limited number of subjects studied. This condition does not allow for the adequate evaluation of the relationship between microRNAs and clinical characteristics or their potential prognostic significance.

Widera et al. [141] addressed this concern by evaluating the diagnostic and prognostic usefulness of cardiomyocyte-enriched miRNAs in various clinical contexts. However, their results reignited previous speculation regarding the potential usefulness of cardiomyocyte-enriched miRNAs as diagnostic or prognostic indicators in ACS. The research examined highly sensitive biomarkers of myonecrosis in a larger cohort of 444 ACS patients. Concentrations of MiR-1, miR-133a, miR-133b, miR-208a, miR-208b, and miR-499 were assessed via quantitative reverse transcription polymerase chain reaction system (qRT-PCR) in plasma samples collected from patients upon admission to the ED or those who underwent PCI. A multiple linear regression analysis, which took into account clinical variables and hsTnT, miR-1, miR-133a, miR-133b, and miR-208b, established independent connections with hsTnT levels (all *p* < 0.001). Importantly, in patients who experienced myocardial infarction, there were higher levels of miR-1, miR-133a, and miR-208b observed than in those with unstable angina. However, all six miRNAs investigated exhibited a wide range across subjects with either unstable angina or myocardial infarction. Levels of noncoding RNA sequences, such as miR-133a and miR-208b, were significant predictors for mortality risk in both univariate and age- and gender-adjusted analyses. However, after further adjustment for hsTnT through statistical discrimination, both miRNAs no longer maintained their independent association with mortality outcome [141].

Oerlemans et al. [134] conducted a prospective study on the role of circulating miRNAs in ACS on a large cohort of patients in order to reinforce the results derived from previous studies with a smaller sample size. Additionally, the previous studies did not include comparisons with stability markers of cardiac injury, nor did they have suitable controls, which this study aimed to address. Briefly, 332 patients with suspected ACS admitted to the ED were enrolled for a single-centre evaluation into the potential diagnostic value of circulating microRNAs as novel biomarkers, including cardiac miRNAs (miR-1, miR-208a, and miR-499), miR-21, and miR-146a. Subjects with STEMI were not deemed eligible for enrolment, which included both UA and NSTEMI patients. The levels of all analysed miRNAs significantly rose in 106 patients clinically diagnosed with ACS. The results indicated that miR-NA-208a and miRNA-146a levels were higher in NSTEMI patients in comparison to those with UA. Conversely, miRNA-1 and miRNA-499 levels tended towards a clinical picture of UA and were higher in these patients than those in NSTEMI patients. The levels of miRNA-21 were similar in NSTEMI and UA subjects. Circulating miRNAs detected in NSTEMI and UA patients present significant potential as new early biomarkers for managing individuals suspected of having ACS.

Bai et al. [135] have expanded upon the scope by examining a more varied demographic of individuals with ACS. Using gene chip technology, the authors assessed miRNA expression levels in patients with SA, NSTEMI, and STEMI. Five individuals from each category and five controls without CAD were included. All participants exhibited three or more risk factors. Microarray analysis was implemented to identify discrepancies in miRNA expression levels, which were subsequently confirmed through qRT-PCR. Patients in the control group and those included in the SA or NSTEMI groups showed differentially expressed miRNAs that were involved in inflammation, protein phosphorylation, and cell adhesion, as compared to miRNAs from STEMI patients. Significant elevation of miR-941, miR-363-3p, and miR-182-5p was observed (fold change: 2.0 or greater, *p* < 0.05) in the control, AS, or NSTEMI, respectively, compared with the expression levels in STEMI patients. Furthermore, qRT-PCR analysis revealed an increase in plasma miR-941 level among NSTEMI or STEMI cohorts in comparison to patients without CAD (fold change: 1.65 and 2.28, respectively; *p* < 0.05). miR-941 expression was significantly increased in the STEMI group when compared to the SA (*p* < 0.01) and NSTEMI (*p* < 0.05) groups. Furthermore, miR-941 expression was higher in patients with ACS and NSTEMI or STEMI, in contrast to patients with SA or without ACS or CAD (*p* < 0.01). These findings suggest that miR-941 expression is relatively higher in patients with ACS and STEMI, potentially indicating its role as a biomarker for ACS or STEMI [135].

Two studies by Wang et al. [136,137] merit comprehensive discussion in the context of circulating miRNAs related to atherosclerosis and support novel and sensitive predictors for acute myocardial infarction. The authors reported changes in the expression of circulating miRNA-21-5p, miRNA-361-5p, and circulating miRNA-519e-5p in patients with coronary atherosclerosis using miRNA microarrays. However, no significant evidence exists to support a global impact of the expression levels of these circulating miRNAs during the initial stage of AMI [26,28].

In the earlier analysis [136], the levels of circulating miR-21-5p, miR-361-5p, and miR-519e-5p were investigated in patients with AMI, and simultaneously, their clinical utility for AMI diagnosis and monitoring was assessed. The study groups comprised the initial cohort of 17 AMI patients and 28 fit volunteers, whereas the second cohort consisted of 9 AMI patients, 9 patients suffering from ischaemic stroke, 8 pulmonary embolism patients, and 12 healthy volunteers. Quantitative real-time PCR and enzyme-linked immunosorbent assay (ELISA) assays were conducted to ascertain plasma miRNA levels and cTnI concentrations in the blood, respectively. The findings indicated that patients with AMI exhibited elevated levels of miR-21-5p and miR-361-5p in their plasma, while the concentration of circulating miR-519e-5p was reduced. Notably, the observed increase in the levels of these circulating miRNAs was strongly correlated with plasma cTnI concentrations. After evaluating the receiver operating characteristics (ROC), we observed a remarkable diagnostic accuracy of these three circulating miRNA for AMI, demonstrating high values of area under the ROC curve (AUC). Therefore, it became apparent that the combination of the three miRNAs significantly increased the diagnostic precision. In this report, Wang et al. established that circulating miRNAs could be regarded as novel and potent biomarkers for AMI, representing a prospective diagnostic tool for the condition [136].

In the study that followed, Wang et al. [137] focused on investigating the link between circulating miRNAs and high-risk characteristics in patients with NSTEMI, utilising a GRACE (Global Registry of Acute Coronary Events) analysis. RNA was extracted from the whole blood of 199 patients with NSTEMI, and whole-genome miRNA sequencing was conducted. To verify the study’s reliability, 13 high-risk clinical traits were examined using generalised linear models. In this regard, the GRACE risk score has been extensively validated for mortality among patients with NSTEMI. A total of 205 miRNA risk factor associations were found to be nominally significant (*p* < 0.05). Upon eliminating the false discovery rate of 5%, it was observed that chronic heart failure had a substantial association with lower levels of circulating miR-3135b (*p* < 0.0006), miR-126-5p (*p* < 0.0001), miR-142-5p (*p* = 0.0004), and miR-144-5p (*p* = 0.0007).

Kaur et al. [138] assessed dysregulated miRNA biomarkers in CAD through the screening of 140 original articles that presented adequate evidence for data mining. Their systematic review collated data from these studies to compare miRNAs identified in patients with ACS, matched with stable CAD patients and control groups. The most frequently indicated miRNAs in any CAD were miR-1, miR-133a, miR-208a/b, and miR-499, which were also abundantly noted within the heart. The authors noted that certain miRNAs were consistently identified in multiple studies, along with their expression levels in either the ACS group or the stable CAD group, as opposed to the control group. However, certain miRNAs have been recognised as biomarkers specifically in ACS patients who demonstrate plasma levels of miR-499, miR-1, miR-133a/b, and miR-208a/b and in stable CAD patients with plasma levels of miR-215, miR-487a, and miR-502. Therefore, elevated plasma levels of miR-21, miR-133, and miR-499 seem to suggest greater potential as biomarkers to differentiate the diagnosis of ACS from stable CAD. Particular attention should be given to miR-499, which has shown a correlation between the slope of its concentration and myocardial damage. It is worth noting that while these miRNAs may offer guidance towards potential diagnostic biomarkers, the reported findings suggest caution in their interpretation. Concerns arise from the implementation of most studies, which are based on candidate-driven, predetermined assessments of a restricted number of miRNAs.

Recently, Zhelankin et al. [139] conducted a study on circulating microRNAs as noninvasive biomarkers of cardiovascular disease in CAD and ACS. The authors highlighted concerns over the controversy and inconsistency of data reported for certain miRNAs, probably due to preanalytical and methodological discrepancies that arose in various studies. The research sample consisted of 136 adult participants who experienced CAD or ACS clinical onset, which included NSTEMI and STEMI admitted patients. Controls included outpatients who were either healthy or had hypertension but not CAD. Patients with ACS had significantly higher plasma levels of miR-21-5p and miR-146a-5p, whereas the level of miR-17-5p was lower in individuals presenting with ACS and stable CAD than in the control group, which consisted of healthy or hypertensive patients without CAD, respectively. Within the ACS patient group, there were no significant differences in the plasma levels of these miRNAs between patients with positive and negative troponins, nor were there any variances found between STEMI and NSTEMI. These findings suggest that raised plasma concentrations of miR-146a-5p and miR-21-5p could be considered as general circulating biomarkers for ACS, while decreased miR-17-5p may be considered a typical biomarker for CAD. No differences in the plasma concentrations of these miRNAs were found among patients in the ACS group, regardless of their troponin levels, or between patients with STEMI and NSTEMI. It is therefore evident that elevated plasma levels of miR-146a-5p and miR-21-5p can be identified as universal biomarkers for ACS, while decreased levels of miR-17-5p can be regarded as a universal biomarker for CAD.

## 6. Implication of miRNA in In-Stent Restenosis

Although fascinating evidence concerning the involvement of miRNAs in in-stent restenosis has only recently been disclosed, we feel it necessary to explore this specific action in greater detail. Changes in miRNA levels have been assessed in cases of in-stent restenosis, a complication that arises after PCI. Both bare metal stents and drug-eluting stents can exhibit anomalous intra-stent neoproliferation. However, with bare metal stents, the manifestation of vascular lesions promotes uncontrolled neointimal formation and subsequent intra-stent restenosis due to the occurrence of atypical proliferation and smooth muscle cell migration [142,143,144]. While drug-eluting stents have mitigated tumour proliferation, the desired significant reduction in the risk of restenosis within the stent is not always achieved. However, the greater risk of late thrombosis associated with drug-eluting stents is a concern that must be addressed. The promotion of late thrombosis in these stents is attributed to the regenerative ability of endothelial cells resulting from vascular wall injury and the prolonged arterial healing process [145,146,147,148,149].

The modulation of miRNA levels and the functionality of such miRNAs, which are facilitated by the implantation of either bare metal or drug-eluting stents, have been examined in pig, mouse, and in vitro models [91]. Firstly, in a porcine animal model, levels of miRNA were analysed in control nonstented and stent-receiving coronary arteries. The findings indicated the overexpression of various pro-inflammatory miRNAs, notably miRNA-21, in stented coronary arteries. Secondly, stent miRNA-21 knockout (KO) mice and stent wild-type mice were compared. The stented miRNA-21 knockout (KO) group experienced a decrease in neointima-medial ratio, neointimal thickness, and neointimal area compared to the stented wild-type control group. Further analysis indicated a higher percentage of anti-inflammatory M2 macrophages in the miRNA-21 KO group compared with the wild-type control group.

The study’s major finding supported a decrease in neointimal proliferation in miRNA-21 stented (KO). This resulted from a significant shift towards M2 macrophage differentiation linked to decreased smooth muscle cell proliferation and migration while preserving endothelial cell function. The results imply that miRNA-21 stimulates inflammation and vessel remodelling following stent implantation. Therefore, the following step could involve experimenting with and subsequently applying medication that controls the expression and plasma levels of miRNA-21, which may enhance the effectiveness of the current drug-eluting stents used to treat thrombosis [91].

Wang et al. [150] investigated the identified role of microRNAs in the intertwined process of myointimal hyperplasia/in-stent restenosis through a humanised animal model. Of the candidates proposed in the study, only miR-21 was found to be upregulated. Additionally, an increase in miR-21 expression was observed in human tissue samples collected from patients with in-stent restenosis compared with those with coronary heart disease. Again, miR-21 was systematically inhibited using intravenous fluorescein-blocked nucleic acid–anti-miR-21 (anti-21) in the humanised myointimal hyperplasia model employed. As expected, the decrease in vascular miR-21 was closely associated in a dose-dependent manner, resulting in reduced occlusion of the lumen. Additionally, anti-21 did not impede the re-endothelialisation process. However, administering anti-miR-21 systemically led to off-target adverse effects, indicating lowered expression of miR-21 in the liver, heart, lungs, and kidneys along with elevated serum creatinine levels. The efficacy of local suppression of miR-21 was evaluated by employing anti-21-coated stents. The results demonstrated that anti-21-coated stents effectively reduced in-stent restenosis compared with bare metal stents. Furthermore, there were no significant off-target effects observed. These findings suggest that anti-miR-coated stents may offer a viable therapeutic option for reducing in-stent restenosis [150].

More recently, Wang et al. [151] have developed a cardiovascular stent, which delivers miRNA to regulate SMCs. The stent is designed using miR-22 as the template miRNA, and it functions through the self-healing encapsulation process based on an amphipathic triblock copolymer spongy mesh made of poly(ε-caprolactone)–poly (ethylene glycol)–poly(ε-caprolactone) (PCL-PEG-PCL, PCEC). In this study, the continuous release of miR-22 via stenting notably enhanced the contractile properties of SMCs, without any hindrance to EC proliferation. Consequently, it resulted in predominant EC growth, with an EC/SMC ratio of 5.4. Wang et al.’s research achieved a significant leap by unveiling how PCEC@miR-22-coated stents minimised inflammation, prevented excessive morphological changes, and inhibited the secretion of extracellular matrix, leading to a significant reduction in in-stent restenosis. This discovery forms the foundation for developing a concise and sturdy coating platform for the transportation of miRNA on cardiovascular stents. This innovation can be expanded to other medical devices and promote the practical implementation of bioactive agents in medical centres (Figure 5).

## 7. Limitations

Of the 25 clinical trials examining microRNAs with therapeutic potential in cardiovascular disease, 17 have completed phase I or II. Although some trials reported withdrawals, those that had completed the clinical phase and targeted microRNA for specific indications, such as miravirsen, RG-101, cobomarsen, and AZD 4076, including the impact in the cardiovascular field with miR-132-3p/inhibitor-CDR132L, showed less promise than anticipated [22,97,98,152,153,154,155,156,157,158,159]. However, it is important to emphasise that preclinical and clinical data obtained with inhibitory oligonucleotides—including clinical trials that have been discontinued—provide valuable information for designing and performing microRNA-targeted cardiovascular therapies. Special attention is paid to miR-17-5p, miR-21-5p, miR-29b-3p, and miR-92a-3p, which have undergone extensive laboratory and clinical study. A notable mention is the miR-132-3p (CDR132L) inhibitor formulated to treat heart failure [9]. Significant strides have been made in studying CDR132L, and the planning of phase II trials is underway. With regard to cardiovascular therapies, this drug could be the first to target miRNAs [9,22,25,101].

## 8. Conclusions

We examined the function of miRNAs in the domain of cardiovascular disease and analysed numerous clinical studies directed towards microRNAs. The increasing amount of clinical studies that focus on microRNAs, culminating in the first clinical study of an anti-miR in cardiovascular therapy, is a definite testimony to the progress achieved over the past decade. The existence of numerous uncharacterised microRNAs suggests that the scope of disease conditions and potential applications of microRNA therapeutics is wider than currently apparent.

Although significant progress has been made in the last decade, indicating the need to take a decisive step towards innovative approaches for diagnosing and treating cardiovascular diseases, there is still a lack of clear differentiation among the results of many microRNAs. It is conceivable that a broader range of disease conditions and applications of microRNA therapies will emerge from the evaluation of the current evidence. The literature reviewed here is pertinent. Further investigation into microRNAs is necessary [159] to authenticate the role of microRNA candidates in cardiology. This will be paramount in considerably enhancing the progress of therapy development and curbing the likelihood of attrition. Thus, we anticipate favourable outcomes from the manipulation of microRNA in cardiovascular disease models alongside omic technologies and extensive preclinical trials.

With the progress achieved in synthesising oligonucleotides, it appears that some big obstacles have been tackled. Nonetheless, delivering these molecules still poses a substantial challenge that necessitates further research. This is particularly relevant to cardiovascular tissues, which inefficiently absorb oligonucleotides. Another important resolution for the administration of potential drugs is the specific modification of oligonucleotides to improve both their absorption and cellular specificity. This aspect of pharmaceutical research is currently not very advanced, and there is considerable uncertainty surrounding the optimal concentrations of cellular oligonucleotides. This implies that significant efforts will be be necessary to screen ligands and chemically link them to oligonucleotides, followed by their usage.

Finally, the widespread implementation of miRNAs is still experimental, but it shows promising potential in diagnosing, prognosing, and treating various presentations of cardiovascular diseases. Its possible therapeutic application could also be beneficial for the upcoming generation of stents. Its integration, whether alone or in combination with currently available biomarkers, may be adopted in the near future, especially when there is diagnostic uncertainty. Further research is necessary to confirm its suitability in the common clinical environment, with a focus on replicability and precision.

## Figures and Tables

**Figure 1 ijms-24-14277-f001:**
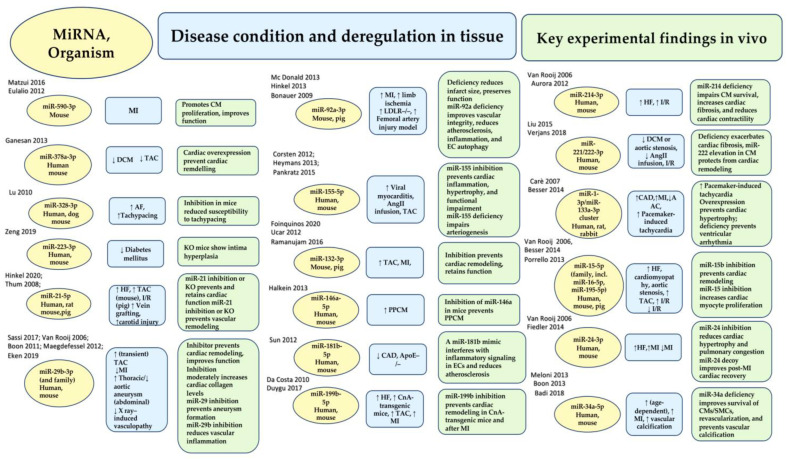
miRNAs have crucial functions in the cardiovascular system. They are involved in disease or disease models and can be manipulated in vivo for their effects. The different boxes in the figure represent the miRNA type and organism evaluated (yellow), the disease condition and regulation (blue), and the key experimental data in vivo (green). Abbreviations: AF, atrial fibrillation; AngII, angiotensin II; CAD, coronary artery disease; CnA, human calcineurin subunit A; DCM, dilated cardiomyopathy; HF, heart failure; I/R, cardiac ischaemia–reperfusion; CM, cardiac myocyte; EC, endothelial cell; KO, knockout; SMC, smooth muscle cell; MI; myocardial infarction; PPCM, peripartum cardiomyopathy; AAC/TAC, ascending/transverse aortic constriction. Refs. [7,8,9,10,11,12,13,14,15,16,17,18,19,20,21,22,23,24,25,26,27,28,29,30,31,32,33,34,35,36,37,38,39,40,41,42,43,44,45,46,47,48] in the figure.

**Figure 2 ijms-24-14277-f002:**
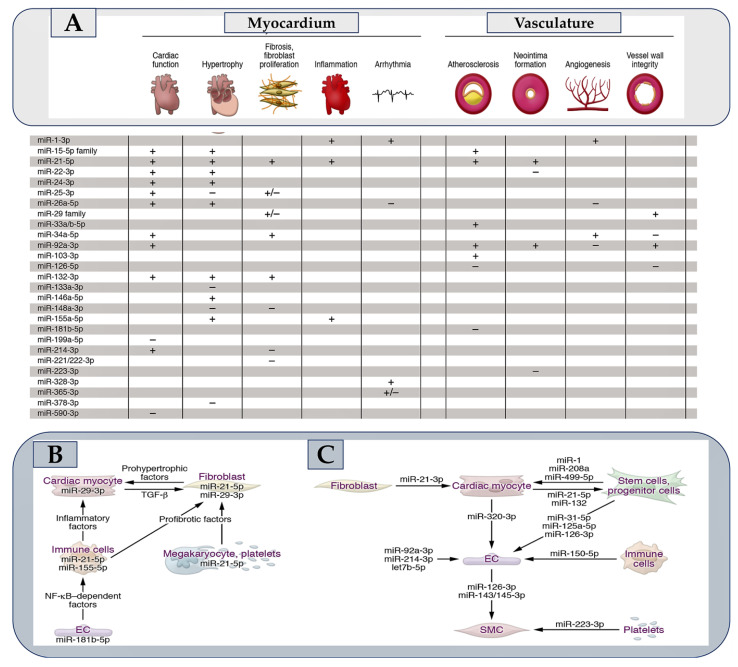
In section A, we provide a summary of the roles of miRNAs in the heart muscle and blood vessels. The suggested miRNA is indicated by a + sign, reflecting its promotion of a process. On the other hand, – sign denotes the prevention of a pathophysiological process by the suggested miRNA. The corresponding columns contain information on the microRNAs that either enhance or impede cardiac function when their levels are elevated or inhibited. Section B outlines instances of microRNAs that control targets responsible for intercellular communication throughout the cardiovascular system. Section C details the paracrine roles of miRNA examples secreted within the cardiovascular system. No amendments are necessary as the text meets the principles, and context is not present. The miR-21 passenger strand (3′) is found in higher concentrations in exosomes from cardiac fibroblasts, leading to cardiac myocyte hypertrophy. Conversely, the miR-21 lead strand, released by endometrial mesenchymal stem cells, provides cardioprotective effects, promoting cell survival and angiogenesis. In contrast, several miRNAs originating from the myocardium stimulate the mobilisation of progenitor cells in the bone marrow. Platelets carry miR-223-3p, which controls the differentiation and proliferation of vascular SMCs. For a summary of different cardiovascular microRNAs with suggested paracrine activity, see Ref. [3]. Abbreviations: EC; endothelial cell; miRNA, microRNA; SMC, smooth muscle cell. From Laggerbauer B et al. [3,7,8,9,10,11,12,13,14,15,16,17,18,19,20,21,22,23,24,25,26,27,28,29,30,31,32,33,34,35,36,37,38,39,40,41,42,43,44,45,46,47,48,52,53,54,55,56,57,58].

**Figure 3 ijms-24-14277-f003:**
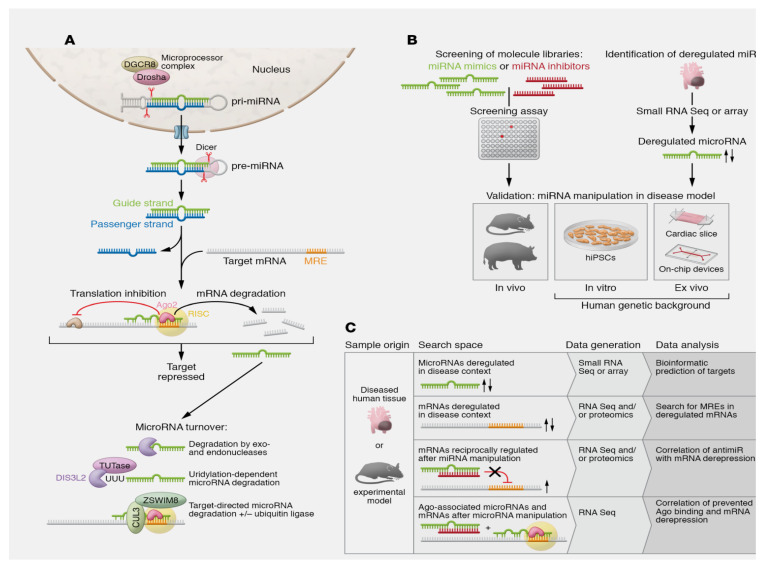
The biogenesis and function of miRNA are reported. Nuclear pre-miRNAs are synthesised and released into the cytoplasm where three main steps promote the final synthesis of activated RNAs that progress alongside the production of miRNA duplex, RISC complex, and RNAi: (**A**) Canonical elaboration, functional activation, mechanism of action, and degradation pathways of microRNAs are reported. The first step of canonical miRNA biogenesis initiates from greater hairpin RNA molecules (pre-miRNAs), which are produced through RNA Pol II transcription of miRNA genes or clusters, or which occur as part of introns. The second step involves a microprocessor complex restraining the endonuclease Drosha, the DGCR8, and other factors, before cleaving the pre-miRNAs. Finally, the resulting pre-miRNA is exported to the cytoplasm, where the nuclease Dicer works to tailor from 21 to 22 nucleotides in length. Noncanonical mechanisms of miRNA biogenesis are also described, some of which bypass the microprocessor complex or Dicer. Following processing into a duplex of 21–22 nucleotides in length each, one strand, which is called the leader strand, becomes part of the RISC, while the passenger strand undergoes accelerated degradation. In case both strands are maintained, individual functions can be adopted, as demonstrated for cardiovascular miR-21 and miR-126. Another exception is miRNA strands that localise to the nucleus, where they function in unusual ways. The degradation step of miRNAs includes the role of the exonucleases XRN-1, PNPase old-35, and RRP41 or the endonuclease Tudor-SN. The DIS3L2 nuclease degrades a subset of miRNAs after modification by TUTases. Mechanisms of TDMD have been resolved, including the involvement of ubiquitin ligases. (**B**) Pathways towards the identification and validation of disease-relevant cardiovascular miRNAs are reported. (**C**) Approaches for identifying microRNA targets are depicted. Abbreviations; DGCR8, DiGeorge critical region 8 protein; DIS3L2, DIS3 like 3′–5′ exoribonuclease 2; miRNA, microRNA; miRNA duplex, precursor miRNA; RISC complex, RNA-induced silencing complex; RNAi, RNA activation; TDMD, target-directed microRNA degradation; TUTases, terminal uridyltransferases. From Laggerbauer B et al. [3,52,70,78,79,93,94,95,96].

**Figure 4 ijms-24-14277-f004:**
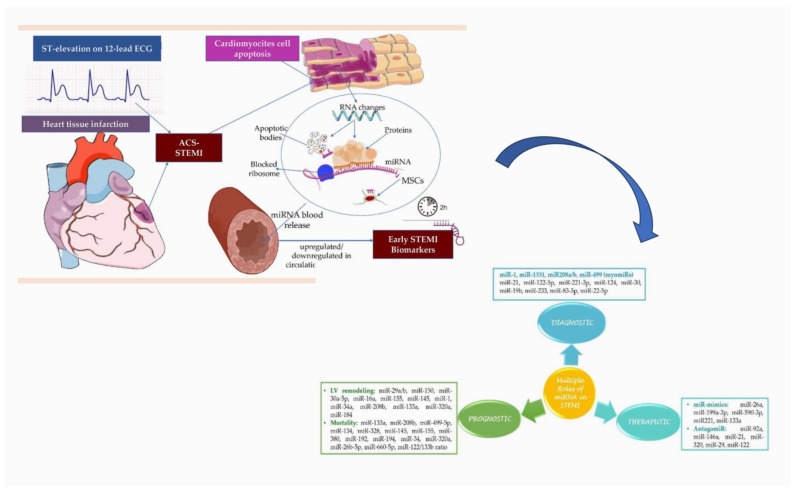
**Up.** Delivery of miRNAs into the bloodstream is facilitated during STEMI as a response to impaired cardiomyocytes. These cells release circulating miRNAs via protein complexes, microvesicles, exosomes, apoptotic bodies, and/or mesenchymal stem cells, resulting in the upregulation or downregulation of miRNAs. **Down.** These miRNAs, particularly downregulated miRNAs, may serve as a diagnostic, prognostic, and therapeutic pathway for patients with STEMI. Abbreviations: ST, ST Segment; ECG, electrocardiogram; ACS, acute coronary syndrome; STEMI, ST-segment elevation myocardial infarction; RNA, ribonucleic acid; miRNAs, microRNAs; MSCs, mesenchymal stem cells. From Tanase DM et al. Life (Basel). [118] and Scărlătescu AI et al. Int J Mol Sci. [121].

**Figure 5 ijms-24-14277-f005:**
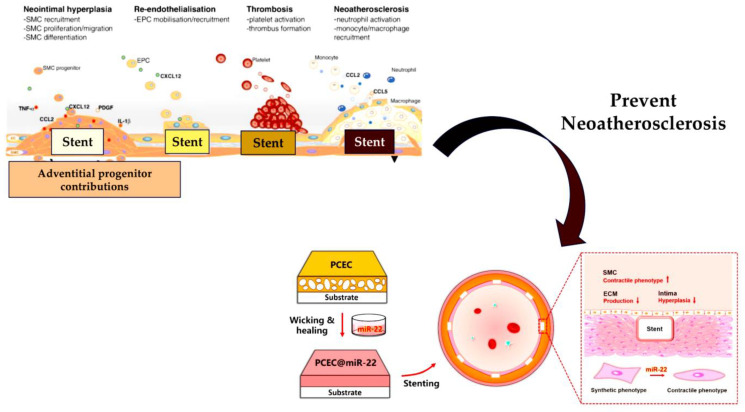
Schematic illustration of the preparation of the PCEC@miR-22-coated stent. The miR-22 was loaded into the PCEC porous coating via a wicking action, followed by self-healing encapsulation. From Wang et al. [150].

## Data Availability

Not applicable.

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
