# Peer review of "The Roles of microRNAs in the Cardiovascular System"

_ijms, 2023, doi:10.3390/ijms241814277_

Round 1
Reviewer 1 Report
In the current review paper Nappi et al. provide an overview of the roles microRNAs may play in cardiovascular diseases. They aim to explore the current state of knowledge about microRNAs as potential targets for diagnosis and therapy in the cardiovascular system.
Providing a comprehensive review on such a broad topic is quite a challenge with a big risk for being incomplete in the overview (which indeed in the case in this review, despite its lengthy descriptions), apart from the fact that already so many overview papers on this topic exist. Therefor my general advise would be to focus this review on one specific subtopic, and limit the information provided only to that specific topic. And thus avoid to try to discuss the complete field of microRNA functions in cardiovascular diseases.
A suggestion might be that they only focus on biomarkers, or miRs that have been translated in the drug in clinical applications. Covering everything is simply too broad.
Comments:
Line 42-43: the remark that one 150 miRNA may have a critical role in CVD comes completely out of the blue and is not supported by any data or references.
Line 80-82 : “ The aim of this review is to highlight the action of some of these candidate miRNAs 80 in relation to their cardiovascular role and their potential clinical development in the di- 81 agnosis and therapy of cardiovascular diseases” : if this the real aim of the review paper , please state this in the title and abstract too.
104-126 : this section on the general principle of microRNA biogenesis and functioning may be shortened extensively, and the figure 2a is quite redundant
Section 3: “Searching roles of microRNAs in the cardiovascular system”
This section seems have a quite strong cherry-picking feeling. Why these microRNAs ? please provide a better rationale for describing these miRNAs and theirs relation to CVD. E.g. the most prominently expressed, most described in literature, uniquely linked to CVD, etc. Now it is suggested that these example miRs are randomly picked, while there are so many more out there that are linked to CVD and not mentioned.
Line 260-282 : this description of acute coronary syndrome and complications may be shortened substantially
Figure 3 : the text in the figure should be in a larger font-size, now large parts are hardly readable
Line 320-420 : this description of inconsistent data on microRNAs with potential biomarker function should be compacted, and to my opinion be summarized as: no consistent biomarker pattern can be defined . But that is my opinion, the authors at least should provide a conclusive paragraph to the section, and not only provide a sum-up of the studies reported here.
Section 5.2 : this summing up of studies is non-informative, and difficult to comprehend. The authors should provide their expert opinion as a recognizable conclusion paragraph to this section.
Section 6 the complete section on in-stent restenosis may be skipped since it is solely focused on miR21.
Figure 4 is not mentioned or discussed in the text of the manuscript and should be removed
no specific comments
Author Response
The authors thank the reviewer for comments.
The manuscript was revised as suggested by the reviewer.
As regards section 5.2, it has been modified. We believe that section 6 is very important and the role played by miRNAs in restenosis is crucial. Although the available data reported are limited they may be of interest to readers
Reviewer 2 Report
The review article titled "Roles of microRNAs in the cardiovascular system" by Nappi et al., is quite extensive. It is well written and the Figures are appropriate. The levels of microRNAs in various disease conditions alter are can be as a result of etiology of the disease or it be used as a marker for the diagnosis. Various relevant studies have been discussed in the review and brings out the importance of microRNAs in cardiovascular disease.
Minor comments:
1) Line 46. "Figure 1" hangs in the air and is unable to place it with the previous sentence of the forthcoming sentence.
2) Line 73, Figure 1 A is not in the article. The sentence seems broken. Correct it.
3) line 83, Table 1 is not there in the article. Either refer to this correctly in the article or remove it
4) Line 79, Figure 1 and graphical abstract again hangs in the air. Please make the references to Figures in the text more in context with the sentences saying the things in the Figure. Some places are fine but some places are confusing.
5) Line 161, expand ACS. Expand abbreviations at first use.
6) In Figure 3, increase the font size in the text inside figure boxes in the below panel. In the print out version of the article it is not clearly visible.
7) In Figure 4, also increase the font size of text inside the figures.
Minor corrections regarding the placement of references to the Figure number may be done.
Author Response
The authors thank the reviewer for comments.
The manuscript was revised as suggested by the reviewer
Reviewer 3 Report
Before I review the whole manuscript, I often look at figures first to make sure that all figures are original and in good quality. I find that critical parts of figures are copies of published reports by other investigators, and I do not see permissions granted by these authors. The images I did identify as copies are: the top part of Fig. 2A. This illustration is from Ingelson-Filpula and Storey, "MicroRNA biogenesis proteins follow tissue-dependent expression during freezing in Dryophytes versicolor". J. Compara. Physiol. B. (2022) 192:611; and the top part of Fig. 3. This comes from Fig. 1 of Tanase et al. "Current knowledge of microRANs (MiRNAs) in acute coronary syndrom (ACS): ST-elevation myocardial infarction (STEMI)". Life (2021) 11(10), doi.org/10.3390/life11101057. This is an MDPI journal. Although I stopped searching for more copy-and-paste images (both large and small), there may be more. This is a form of misconduct in science. The authors should withdraw this paper immediately.
Need editing
Author Response
The authors thank the reviewer for comments.
The possibility of reusing already published figures is allowed either because they have been published under a Creative Commons 4.0 attribution license or by accessing the Copyright Clearance Center site.
For all the figures that have been included in the manuscript, reuse has been requested.
Figure 1 is original.
Figure 2 was requested to be reused by JCI. The editor's response is included
Hi Francesco,
These figures were published under a Creative Commons 4.0 attribution
license.
No permission is required, and you are free to reuse or adapt as long as
you cite the original article. Details of terms are below.
https://creativecommons.org/licenses/by/4.0/
Sarah Jackson, PhD
Executive Editor
The Journal of Clinical Investigation
Regarding figure 3 (upper part) the request to reuse the figure was made to MDPI. The editor's response is included
Dear Francesco,
Many thanks for your email. Since Life is an open access journal, please feel free to use the figure.
Have a nice day!
Best,
Lila
Life life@mdpi.com
Therefore there are well-founded requirements to leave the figures in place
Round 2
Reviewer 1 Report
Thank you for adapting the ms and zooming in on the biomarker role.
Please refit figure 4 so it fits into the template. Now the right part, with as I assume the list of miRs involved in restenosis, falls of the figure.
But I still have a serious problem with section 6. In the text you only refer to the work with miR21, anti-miR21 and the miR21 KO mice, and than figure 4 refers to PCEC@miR22. Please describe the role of miR22 in restenosis also in the text of section 6, or skip figure 4 that still is not discussed.
just a few sentences that contain strange grammatical constructions, nothing serious
Author Response
Thank you for adapting the ms and delving into the role of the biomarker.
Reassemble figure 4 to fit your model. Now the right side, with I assume the list of miRs involved in restenosis, falls out of the figure.
But I still have a serious problem with section 6. In the text it only refers to the work with miR21, anti-miR21 and miR21 KO mice, and then figure 4 refers to PCEC@miR22. Please describe the role of miR22 in restenosis al.
The authors thank the reviewer for the comments.
We apologize for the error on the file sent. This revised text reports the recent work by Wang and colleagues (Ref 151) cited in Figure 4
Reviewer 3 Report
General comments.
The authors reviewed correlation between a number of cardiovascular diseases and expression levels of miRs with special emphasis on circulating miRs. The topics covered in this review are sufficient. Unfortunately, the quality of writing is poor, which sometimes makes it challenging to evaluate the scientific contents. Listed below are several specific comments.
1. Due to the style of writing, many statements are ambiguous, unclear, and sometimes misleading. There are several reasons. 1) In many cases, word choices are strange (unconventional) and often inappropriate. 2) Some sentences are long and the sentence structure is complex. 3) Some sentences are “flowery”, which in science, could make a statement ambiguous. The text must be edited by a biologist who understands the content. An English editor who does not fully understand the content CANNOT fix all the language issues.
2. If a certain figure was borrowed from papers published by others, please give credit to them by citing the paper in the legend or in acknowledgements.
3. Define circulating miR. It appears that some studies were done using the whole blood and also individual cellular components in peripheral blood. Are circulating miRs different from plasma miRs?
4. The authors state that miRs may be used for the diagnosis, prognosis and therapy of a variety of cardiovascular diseases. I agree that they may be useful for prognosis and treatment, but I am not sure if it is as useful for diagnosis simply because of the timing issue. In many cases of acute cardiac events, one has little time to waste. MiR assays which take a few hours may not be feasible. The currently available troponin tests take as short as a few minutes. The authors may wish to discuss this issue, which is specific to CVD, not to cancer and other non-acute illnesses.
5. Figure 1 is a useful summary. It would be good if the authors could include references (by number) in the figure so that readers do not have to spend time to look for them.
6. May abbreviations are defined in a very haphazard manner. Please follow the instruction of the journal. By the way the Center for Disease Control and Prevention is CDC, not CDS.
7. Lines 386-387. ”Instead, miR-133a levels…..although the reported values did not suggest significance.” If the statistics says “not significant”, there is no difference. Please delete this sentence.
8. Lines 633-635. “Although with the advent of drug-eluting stents, tumors develop less frequently, a significant reduction in the risk of restenosis within the stent is less often desired.” This is a perfect sentence, but I have no idea what the authors wish to say.
The text MUST be edited by a BIOLOGIST WHO UNDERSTAND the content.
Author Response
The authors thank the reviewer for the comments
1. Due to the style of writing, many statements are ambiguous, unclear, and sometimes misleading. There are several reasons. 1) In many cases, word choices are strange (unconventional) and often inappropriate. 2) Some sentences are long and the sentence structure is complex. 3) Some sentences are “flowery”, which in science, could make a statement ambiguous. The text must be edited by a biologist who understands the content. An English editor who does not fully understand the content CANNOT fix all the language issues.
The text has been extensively revised
2. If a certain figure was borrowed from papers published by others, please give credit to them by citing the paper in the legend or in acknowledgments.
Figures have been checked and citations added
3. Define circulating miR. It appears that some studies were done using the whole blood and also individual cellular components in peripheral blood. Are circulating miRs different from plasma miRs?
The dosage of miRNAs depends on the study protocol performed. recently, for greater reliability, plasma dosage has been performed
4. The authors state that miRs may be used for the diagnosis, prognosis and therapy of a variety of cardiovascular diseases. I agree that they may be useful for prognosis and treatment, but I am not sure if it is as useful for diagnosis simply because of the timing issue. In many cases of acute cardiac events, one has little time to waste. MiR assays which take a few hours may not be feasible. The currently available troponin tests take as short as a few minutes. The authors may wish to discuss this issue, which is specific to CVD, not to cancer and other non-acute illnesses.
Chapter 5 has been completely revised
5. Figure 1 is a useful summary. It would be good if the authors could include references (by number) in the figure so that readers do not have to spend time to look for them.
References have been added
6. May abbreviations are defined in a very haphazard manner. Please follow the instruction of the journal. By the way the Center for Disease Control and Prevention is CDC, not CDS.
Abbreviations have been revised in the text.
7. Lines 386-387. ”Instead, miR-133a levels…..although the reported values did not suggest significance.” If the statistics says “not significant”, there is no difference. Please delete this sentence.
The text has been revised
8. Lines 633-635. “Although with the advent of drug-eluting stents, tumors develop less frequently, a significant reduction in the risk of restenosis within the stent is less often desired.” This is a perfect sentence, but I have no idea what the authors wish to say.
The text has been revised
The text MUST be edited by a BIOLOGIST WHO UNDERSTAND the content.
Dr. Francesca Bellomo cited in the Acknowledgment, an expert in cardiac biology and genetics revised the text
Round 3
Reviewer 1 Report
the authors have clarified my last remaining issues and I have not further remarks , except that I want to compliment the authors with this excellent review paper
Author Response
The authors are very grateful to the reviewer
Reviewer 3 Report
General comments
The revised MS reads better, but there are still a nuber of language issues, some of which are listed below. Others may be edited by the language editor. Although the authors responded adequately to many of my comments, they either did not understand my comments or they may have decided to ignore them. Such comments are repeated as specific comments below.
Specific Comments
1. Figure 1. I suggested in my previous review to add references to this figure. The authors added “Ref. 7-48” at the end of the figure legend. This is inadequate and unkind in my view. The reason for the reference is so that readers can go directly to the source of information without having to check 40+ references. So, please indicate references in the figure by number; for instance, in the oval yellow area. This will increase the scientific value of this figure.
2. Lines 185-187. “Although a worldwide lack of miR-21-5p goes unnoticed, the reiteration of the inhibitors' impact is shown through a genetic knockout of miR-21 in non-myocyte cells.” I am not sure if I understand this statement. What do you mean by “worldwide lack”? Perhaps, you mean global or systemic OK? Re-write the sentence.
3. Lines 198-203. “A significant study by von Roji et al. demonstrated that miR-29 mimics had a repressive effect on collagen and resulted in an improvement in cardiac function. (15) A notable study conducted by von Roji et al. showed that the use of miR-29 mimics had a suppressive impact on collagen, ultimately leading to enhanced cardiac functionality. (15) Additionally, this study highlighted that miR-29 mimics have the potential to improve cardiovascular health.” This is highly repetitive. Re-write.
4. Lines 238-239. “but a phase II study was halted due to speculative interest in cardiovascular research.” What does this statement mean? Was the study halted due to a non-scientific reason? Please explain fully if this is an important piece of information.
5. Line 324. “….with UA (p > 0.05).” Shouldn’t it be (p <0.05)?
6. Lines 367-368. “Conversely, miR-133a and miR-208a levels were elevated, although the reported values did not indicate significance.” I objected to this statement in my previous review, but the authors did not understand my comments. This statement made by the authors violates the most basic rule of statistics. If a statistic analysis of any kind gives a not-significant result, one MUST conclude there is no difference, in this case no elevation. Delete the statement.
7. Lines 394-411. This paragraph is muddy and confusing. Streamline the description by describing only the important points. Readers can always go to the cited paper for detail. Re-write.
8. Lines 533-535. “However, there was no substantial evidence for a worldwide role of the expression levels of these circulating miRNAs in the initial stage of AMI.” I do not understand this statement. Do you mean expression of certain miRs has an international role? Or do you mean these circulating miRs have some functional roles in (patients living in) certain regions of the world? If so, this is interesting, indeed. Please explain more fully and add references
9. Lines 687-589. “A notable mention is the miR-132-3p (CDR132L) inhibitor formulated to treat heart failure. (9) Significant strides have been made in studying CDR132L, and the planning of phase II trials is underway.” Delete as these statements are repeats.
10. Lines 693-695. “In our research, noteworthy outcomes were highlighted in a published seminal article that resulted in the therapeutic use of an antimiR via inoculation.” I do not clearly understand this statement. It sounds like the authors have published a seminal paper earlier describing their own research work. If so, please cite your seminal paper here. Perhaps, you can re-write the sentence in a clearer manner.
11.The use of abbreviations is still haphazard. This was pointed out in the previous review. Many abbreviations are not defined when used for the first time, some are defined repeatedly, and abbreviations are not always used. The list of abbreviations is incomplete (i.e. many of them are not listed).
Needs careful editing, especially abbreviations, use of the word "respectively", punctuation, and grammar.
Author Response
The authors thank the reviewer for the comments
Figure 1. I suggested in my previous review to add references to this figure. The authors added “Ref. 7-48” at the end of the figure legend. This is inadequate and unkind in my view. The reason for the reference is so that readers can go directly to the source of information without having to check 40+ references. So, please indicate references in the figure by number; for instance, in the oval yellow area. This will increase the scientific value of this figure.
We have modified the references in Figure 1 as suggested by the reviewer
Lines 185-187. “Although a worldwide lack of miR-21-5p goes unnoticed, the reiteration of the inhibitors' impact is shown through a genetic knockout of miR-21 in non-myocyte cells.” I am not sure if I understand this statement. What do you mean by “worldwide lack”? Perhaps, you mean global or systemic OK? Re-write the sentence.
The sentence has been rewritten.
Lines 198-203. “A significant study by von Roji et al. demonstrated that miR-29 mimics had a repressive effect on collagen and resulted in an improvement in cardiac function. (15) A notable study conducted by von Roji et al. showed that the use of miR-29 mimics had a suppressive impact on collagen, ultimately leading to enhanced cardiac functionality. (15) Additionally, this study highlighted that miR-29 mimics have the potential to improve cardiovascular health.” This is highly repetitive. Rewrite.
The sentence has been rewritten.
Lines 238-239. “but a phase II study was halted due to speculative interest in cardiovascular research.” What does this statement mean? Was the study halted due to a non-scientific reason? Please explain fully if this is an important piece of information.
The sentence has been rewritten.
Line 324. “….with UA (p > 0.05).” Shouldn’t it be (p <0.05)?
Changed
Lines 367-368. “Conversely, miR-133a and miR-208a levels were elevated, although the reported values did not indicate significance.” I objected to this statement in my previous review, but the authors did not understand my comments. This statement made by the authors violates the most basic rule of statistics. If a statistic analysis of any kind gives a not-significant result, one MUST conclude there is no difference, in this case no elevation. Delete the statement.
Both statistical significance and non-significance are reported in the results of a scientific study. The sentence has been rewritten.
Lines 394-411. This paragraph is muddy and confusing. Streamline the description by describing only the important points. Readers can always go to the cited paper for detail. Re-write.
The paragrapf has been rewritten. (Line 393-404)
Lines 533-535. “However, there was no substantial evidence for a worldwide role of the expression levels of these circulating miRNAs in the initial stage of AMI.” I do not understand this statement. Do you mean expression of certain miRs has an international role? Or do you mean these circulating miRs have some functional roles in (patients living in) certain regions of the world? If so, this is interesting, indeed. Please explain more fully and add references
The sentence has been rewritten.
Lines 687-589. “A notable mention is the miR-132-3p (CDR132L) inhibitor formulated to treat heart failure. (9) Significant strides have been made in studying CDR132L, and the planning of phase II trials is underway.” Delete as these statements are repeats.
Deleted
Lines 693-695. “In our research, noteworthy outcomes were highlighted in a published seminal article that resulted in the therapeutic use of an antimiR via inoculation.” I do not clearly understand this statement. It sounds like the authors have published a seminal paper earlier describing their own research work. If so, please cite your seminal paper here. Perhaps, you can re-write the sentence in a clearer manner.
The sentence has been rewritten.
The use of abbreviations is still haphazard. This was pointed out in the previous review. Many abbreviations are not defined when used for the first time, some are defined repeatedly, and abbreviations are not always used. The list of abbreviations is incomplete (i.e. many of them are not listed).
We have double-checked all the abbreviations
